# ALIGNING ROTATIONAL AND HIERARCHICAL GEOMETRY IN MOLECULAR REPRESENTATION LEARNING WITH PRODUCT-MANIFOLD LATENT SPACES

## ABSTRACT

Learning effective molecular representations requires capturing two fundamental but largely disjoint aspects of the structure of molecules: rotational symmetries in 3D conformations and the hierarchical organization of chemical scaffolds. We introduce a new paradigm of product-manifold representation learning with product-manifold message passing on $SO(3) \times \mathbb{H}^d$, which couples equivariant geometric features with hyperbolic embeddings of chemical hierarchy. Our construction preserves $SO(3)$-equivariance in the geometric channel while enabling curvature-aware aggregation in the hyperbolic channel, with cross-coupling restricted to scalar invariants to maintain symmetry. Unlike prior approaches that fuse equivariant and hierarchical encoders via concatenation or stacking, our method defines message passing directly on the product manifold, yielding a unified representation. We outline how such models could be evaluated on molecular property prediction, scaffold-split generalization, and generative design, and discuss how embeddings in $SO(3) \times \mathbb{H}^d$ provide a natural surrogate space for manifold Bayesian optimization, enabling more sample-efficient discovery of high-value molecules compared to Euclidean BO. Together, these results suggest a principled path toward unifying physical symmetries and chemical hierarchies within a single geometric learning framework.

## 1 INTRODUCTION

In recent years, graph neural networks (GNNs) have become a central tool for molecular machine learning, yet their progress has revealed a persistent gap: no single model simultaneously respects the rigid-body symmetries of 3D conformations and hierarchical organization of chemical scaffolds. This gap is consequential. Equivariant message passing neural networks (MPNNs) excel at encoding physical laws by ensuring predictions remain stable under rotations and translations of a molecule (Thomas et al., 2018; Fuchs et al., 2020; Satorras et al., 2022; Schütt et al., 2021; Batzner et al., 2022). In parallel, hyperbolic embeddings have proven indispensable for capturing scaffold trees and other hierarchical structures that govern generalization across chemical families (Nickel & Kiela, 2017; Ganea et al., 2018a;b; Wu et al., 2021). But models that specialize in one geometry almost always discard the other. As a result, when trained under scaffold-split regimes (Wu et al., 2018), symmetry-preserving networks struggle with distribution shift, while hierarchy-aware networks ignore the conformational physics that determines molecular properties.

Current attempts to fuse these perspectives remain limited. The dominant strategy is late fusion: *equivariant* and *hyperbolic* encoders are trained separately, and their outputs concatenated or stacked only at the readout stage. This approach, however, leaves cross-channel interactions underconstrained and risks leaking non-invariant information into symmetry-sensitive paths. The consequence is a representation that is neither fully equivariant nor truly hierarchy-aware.

In this work, we argue that unification must happen earlier—at the level of message passing itself. We introduce a product-manifold framework that defines message passing directly on $\mathcal{M} = SO(3) \times \mathbb{H}^d$ (or $E(3) \times \mathbb{H}^d$). Within this construction, a geometric channel encodes vectors and tensors that transform predictably under rigid motions, while a hyperbolic channel embeds scaffold hierarchies using curvature-aware aggregation via Fréchet means, implemented in practice

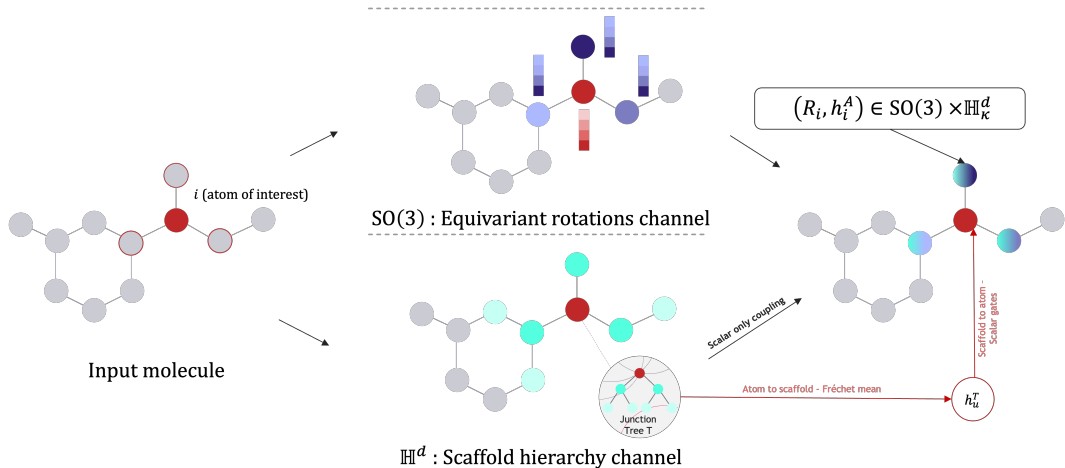

Figure 1: *Conceptual illustration of the product–manifold architecture.* (*Left*) A molecule with a highlighted atom. (*Top*) The SO(3) geometric channel processes 3D neighborhoods using spherical-harmonic edge bases and equivariant tensor products. (*Bottom*) The hyperbolic channel pools atom-level hyperbolic states into scaffold representations via a Fréchet (Lorentz) mean over junction-tree cliques. (*Right*) Scalar-only coupling allows scaffold information to influence geometric message passing through invariant gates (e.g., distances, torsions, hyperbolic similarities) without breaking equivariance.

by Lorentz-mean pooling. Crucially, the two channels communicate only through invariant scalars such as distances, norms, or rotation-invariant statistics. This scalar-only coupling ensures *symmetry safety*: hierarchical information can shape the representation without compromising SO(3)/E(3) equivariance. By pooling and readout on the product manifold, we obtain a single latent geometry where conformational symmetries and scaffold hierarchies are co-encoded by design.

This unified approach matters in three complementary ways. First, symmetry-safe coupling exposes scaffold-level context precisely where equivariant networks fail—under scaffold-split generalization—without sacrificing their physical guarantees. Second, hyperbolic aggregation respects the exponential volume growth of scaffold trees, reducing distortion compared to Euclidean pooling. Third, the hyperbolic factor provides a manifold-appropriate latent space for Bayesian optimization (Calandra et al., 2016; Borovitskiy et al., 2023; Jaquier et al., 2019), enabling intrinsic kernels and acquisition strategies along geodesics. Together, these advances yield a principled framework where molecular physics and chemical hierarchy reinforce one another rather than compete.

We make three key contributions:

1. **Product-manifold message passing:** A molecular GNN operating directly on $SO(3) \times \mathbb{H}^d$, with an equivariant geometric channel, a hyperbolic channel using curvature-aware aggregation, and scalar-only cross-coupling that preserves invariance.

2. **Symmetry-safe integration:** Practical designs for stable manifold operations (projection–retraction updates and Lorentz-mean barycenters) alongside gating constraints that rigorously prevent leakage of orientation-dependent information across channels.

3. **Evaluation protocols:** Benchmarks emphasizing scaffold-split generalization, ablations isolating each design choice, and a demonstration of manifold Bayesian optimization in the latent space as a principled tool for molecular design.

By situating message passing on a mixed-curvature product manifold, our work advances beyond concatenative fusion and provides the first symmetry-safe unification of rotational and hierarchical geometry for molecular learning.

## 2 RELATED WORK

We review equivariant GNNs and transformers, hyperbolic geometry for molecular hierarchies, mixed-curvature spaces, hybrid models combining structural priors, and geometry-aware Bayesian optimization. Our contribution is an *early fusion* architecture on a product manifold, coupling an SO(3)-equivariant geometric channel with a hyperbolic hierarchy channel via scalar-invariant cross-talk. This stands in contrast to late-fusion pipelines that concatenate features without symmetry guarantees.

### 2.1 EQUIVARIANT MOLECULAR GNNS AND TRANSFORMERS

Early message-passing architectures (SchNet, DimeNet, DimeNet++, GemNet) incorporated continuous-filter convolutions and directional interactions to encode distances and angles (Schütt et al., 2018; Klicpera et al., 2020; Gasteiger et al., 2022; 2024). Later models enforced E(3)/SE(3)/SO(3) symmetry through steerable features or constrained coordinate updates: Tensor Field Networks (TFN) (Thomas et al., 2018), SE(3)-Transformer (Fuchs et al., 2020), EGNN (Satorras et al., 2022), PaiNN (Schütt et al., 2021), NequIP (Batzner et al., 2022), SEGNN (Brandstetter et al., 2022), SphereNet (Liu et al., 2022), and more recently Equiformer (Liao & Smidt, 2023) and EquiformerV2 (Liao et al., 2024). Toolkits such as `e3nn` further standardized irreps bookkeeping and gate activations (Geiger & Smidt, 2022).

Recent work has focused on scaling irreps-based attention to higher angular degrees. EquiformerV2 replaces expensive $SO(3)$ tensor products with efficient eSCN spherical-channel convolutions and introduces attention re-normalization, separable $S^2$ activations, and separable layer normalization, yielding state-of-the-art performance on OC20/OC22 Chanussot et al. (2021); Tran et al. (2023) and strong results on QM9 (Liao et al., 2024). Other directions improve efficiency through Cartesian vector features (PaiNN (Schütt et al., 2021)), higher-body-order messages (MACE (Batatia et al., 2023)), or spherical-channel methods (SCN/eSCN (Zitnick et al., 2022; Passaro & Zitnick, 2023)). Our geometric channel is compatible with these families and preserves their equivariance guarantees; our novelty lies in *how* this channel is coupled to a curved hierarchy channel without symmetry leakage.

### 2.2 HYPERBOLIC GEOMETRY FOR MOLECULAR HIERARCHIES

Hyperbolic spaces embed trees with low distortion and support curvature-aware aggregation (e.g., Fréchet means). Foundational work includes Poincaré embeddings (Nickel & Kiela, 2017), hyperbolic neural networks (Ganea et al., 2018a), and HGCN (Ganea et al., 2018b), with theoretical trade-offs between curvature, dimension, and distortion motivating their use (Sa et al., 2018). In chemistry, Bemis–Murcko frameworks and scaffold trees formalize nested chemotypes relevant for scaffold generalization (Bemis & Murcko, 1996; Schuffenhauer et al., 2007). Prior hyperbolic GNNs capture hierarchy but ignore 3D rotational physics; our approach defines message passing directly on a product manifold so that hierarchy and geometry co-evolve.

### 2.3 MIXED-CURVATURE AND PRODUCT SPACES

Product manifolds and mixed-curvature embeddings (e.g., $\mathbb{H}^k \times \mathbb{R}^m \times \mathbb{S}^\ell$) have been applied to knowledge graphs and representation learning, improving fit for heterogeneous relational patterns (Gu et al., 2019; Skopek et al., 2020). These works, however, stop at embeddings or VAEs. We extend the idea to *symmetry-preserving message passing* on $SO(3) \times \mathbb{H}^d$, with cross-channel coupling restricted to invariants.

### 2.4 HYBRIDS THAT COMBINE STRUCTURAL PRIORS

Several models augment equivariance with higher-order geometric structure (angles and dihedrals in DimeNet/GemNet, hypergraph or directional modules in SEGNN) (Klicpera et al., 2020; Gasteiger et al., 2024; Brandstetter et al., 2022). Other pipelines concatenate scaffold descriptors with Euclidean encoders, but such late fusion offers no guarantees against symmetry violation or information leakage. By contrast, our approach performs an *early fusion* within a single product manifold, where SO(3)-equivariance and hyperbolic aggregation are coupled only through scalars.

## 2.5 BAYESIAN OPTIMIZATION BEYOND EUCLIDEAN SPACE

Gaussian processes and Bayesian optimization have been extended to Riemannian manifolds via intrinsic kernels and geodesic acquisitions (Calandra et al., 2016; Borovitskiy et al., 2023; Jaquier et al., 2019; 2023). These methods have proven effective in robotics and control. We advocate Bayesian optimization directly in the hyperbolic (or product) latent space learned by our model, so that acquisition functions respect scaffold hierarchies and geometric invariants—providing a principled alternative to Euclidean BO on equivariant encoders.

## 3 PRELIMINARIES

**Equivariant message passing.** Equivariant message passing enforces that geometric features transform predictably under rotations, allowing the model to process 3D molecular structure without ever breaking SO(3) symmetry. At a high level, features are organized into irreducible representations (irreps) so that a global rotation of the input produces a corresponding, well-defined rotation of all non-scalar channels. Updates to these features are implemented using equivariant linear maps and tensor products that preserve SO(3)-equivariance by construction. For completeness, Appendix A.2 summarizes group actions, irreps, Wigner–$D$ matrices, and Clebsch–Gordan tensor products used in the geometric channel.

**Hyperbolic geometry with projection and retraction.** Hyperbolic spaces embed tree-like structure with low distortion, enabling the hierarchy channel to represent scaffold organization in a geometry that naturally fits molecular scaffold trees. We use the Lorentz (hyperboloid) model of $\mathbb{H}_\kappa^d$, which supports efficient computation of distances, tangent–space projections, and retractions. These operations allow both atom- and scaffold-level states to evolve within a curved space using first-order Riemannian updates. Detailed formulas for the Lorentz inner product, hyperbolic distance, tangent spaces, projection, retraction, and numerical considerations are in Appendix A.3.

**Product-manifold message passing.** Product-manifold message passing couples geometric and hierarchical information by updating atom and scaffold states jointly on $\mathrm{SO}(3) \times \mathbb{H}_\kappa^d$ while preserving the symmetries of each factor. Each atom maintains an SO(3) pose $R_i$, a hyperbolic state $h_i^A$, and irrep features $f_i$, while each scaffold node carries a hyperbolic state $h_u^T$. A single layer performs three operations: (i) an SO(3)-equivariant update of the geometric features, (ii) a Riemannian update of atom- and scaffold-level hyperbolic states, and (iii) a scalar-only coupling in which invariant distances and similarities gate information flow between channels. Because cross-channel communication uses only $\ell = 0$ scalars, the geometric stream remains exactly SO(3)-equivariant.

## 4 METHODS

We begin by outlining the problem and then expand on the main components of our architecture: (i) an SO(3)-equivariant geometric channel processing 3D atomic structure, (ii) a hyperbolic channel encoding scaffold hierarchy, (iii) scalar-only gates that connect the two streams without breaking equivariance, (iv) a joint product–manifold update within each message-passing layer, and (v) an E(3)-invariant readout. Figure 2 provides an overview. This design unifies geometric and hierarchical information in a single symmetry-preserving architecture.

### 4.1 PROBLEM SET-UP AND NOTATION

We are given a molecular graph $G = (V, E)$ with atoms $i \in V$ and bonds $j \to i \in E$, together with 3D coordinates $x_i \in \mathbb{R}^3$ and optional scalar attributes. In addition, we construct a junction-tree scaffold $T = (U, E_T)$ whose nodes $u \in U$ denote ring/bridge cliques and whose edges connect cliques that share atoms or virtual bridge bonds. For each scaffold node $u \in U$, let $S(u) \subseteq V$ be the atoms covered by its subtree, and denote by $u(i) \in U$ the scaffold node containing atom $i$.

Our model maintains, for each atom $i$, an *equivariant geometric channel* consisting of irreducible representation (irrep) features $f_i = \{f_i^{(\ell)}\}_{\ell \in \mathcal{L}}$, where $f_i^{(\ell)} \in \mathbb{R}^{m_\ell \times (2\ell+1)}$ transforms under the

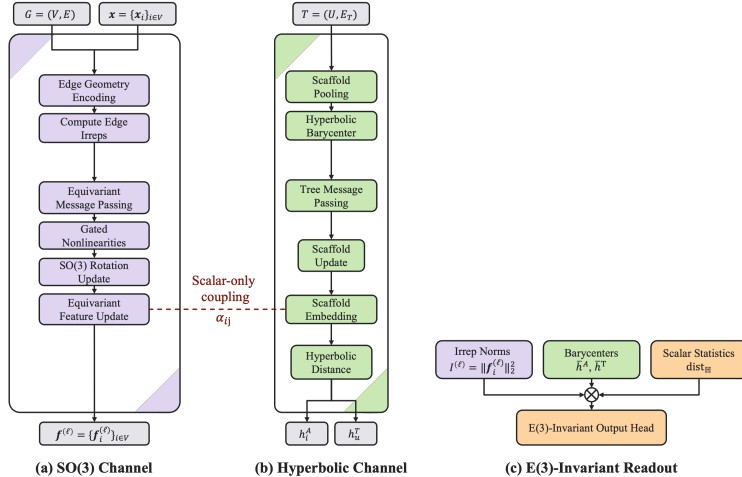

**Figure 2:** Overview of the Product-Manifold GNN architecture. There are two parallel streams: (a) an SO(3)-equivariant geometric channel, and (b) a hyperbolic hierarchy channel. The two channels exchange information exclusively through scalar-only gates $\alpha_{ij}$ (Eq. 10). Both channels contribute invariant summaries—irrep norms, hyperbolic barycenters, and hyperbolic distance statistics—which are merged into a unified (c) E(3)-invariant readout for downstream prediction.

$\ell$-th SO(3) irrep, and a *hyperbolic hierarchy channel* state $h_i^A \in \mathbb{H}_\kappa^d$. Optionally, the model also maintains a node-wise SO(3) pose $R_i \in \mathrm{SO}(3)$. Similarly, each scaffold node $u \in U$ carries a hyperbolic state $h_u^T \in \mathbb{H}_\kappa^d$.

A global rotation $Q \in \mathrm{SO}(3)$ acts as

$$x_i \mapsto Qx_i, \quad R_i \mapsto QR_i, \quad f_i^{(\ell)} \mapsto D^{(\ell)}(Q)\,f_i^{(\ell)}, \quad h_i^A, h_u^T \text{ fixed},$$

where $D^{(\ell)}$ is the real Wigner-$D$ matrix of order $\ell$. All interactions between the geometric and hyperbolic channels use $\ell = 0$ scalar invariants only, so the overall network remains SO(3)-equivariant.

### 4.2 SO(3)-EQUIVARIANT GEOMETRIC CHANNEL

This channel processes 3D atomic geometry using rotation-consistent edge features and equivariant message passing to produce irreps-based updates that respect SO(3) symmetry. For each directed edge $j \to i$, we define

$$r_{ij} = x_j - x_i, \qquad d_{ij} = \|r_{ij}\|, \qquad \hat{r}_{ij} = r_{ij}/d_{ij}.$$

We encode directions using real spherical harmonics $Y^{(\ell)}(\hat{r}_{ij}) \in \mathbb{R}^{2\ell+1}$ and distances using a learnable radial basis $\rho_{\ell n}(d_{ij})$. Any sufficiently smooth 1D basis can be used; in our implementation $\rho_{\ell n}$ is a compactly supported radial basis function implemented in PyTorch (e.g., Gaussian or B-spline), which avoids special functions and is shared across layers.

Edge irreps are formed as

$$e_{ij}^{(\ell)} = \sum_n w_{\ell n}\,\rho_{\ell n}(d_{ij})\,Y^{(\ell)}(\hat{r}_{ij}), \tag{1}$$

where $w_{\ell n}$ are learned scalar coefficients. These features transform in the $\ell$-th irrep and can be re-used by every layer.

We also compute a torsion-like dihedral angle $\varphi_{ij}$ for each edge using four atoms in a local motif; this enters only through even functions (e.g., $\cos\varphi_{ij}$, $\sin^2\varphi_{ij}$) so that scalar targets remain reflection-invariant.

Given these edge features, the geometric channel aggregates information through equivariant message passing. Messages from node $j$ to node $i$ at output order $\ell_o$ are constructed by Clebsch–Gordan

tensor products between neighbour features and edge irreps:

$$m_{i \leftarrow j}^{(\ell_o)} = \sum_{\ell_i, \ell_e} C_{\ell_i, \ell_e \to \ell_o} \big( f_j^{(\ell_i)}, e_{ij}^{(\ell_e)} \big) \, \alpha_{ij}, \tag{2}$$

where $C_{\ell_i, \ell_e \to \ell_o}$ is a fixed linear map assembled from Clebsch–Gordan coefficients, and $\alpha_{ij} \in (0, 1)$ is a scalar gate (see Section 4.4).

Node features are updated via equivariant linear maps and gated nonlinearities:

$$f_i^{(\ell)} \leftarrow \mathrm{Gate}^{(\ell)} \left( W_{\mathrm{self}}^{(\ell)} f_i^{(\ell)} + \sum_{j \in \mathcal{N}(i)} W_{\mathrm{msg}}^{(\ell)} m_{i \leftarrow j}^{(\ell)}, \, s_i \right), \tag{3}$$

where $W_{\mathrm{self}}^{(\ell)}$ and $W_{\mathrm{msg}}^{(\ell)}$ are learnable equivariant linear maps, $s_i$ is an $\ell = 0$ scalar feature, and $\mathrm{Gate}^{(\ell)}$ is a standard e3nn-style gated nonlinearity that uses $s_i$ to modulate non-scalar channels.

### 4.3 HYPERBOLIC SCAFFOLD-HIERARCHY CHANNEL

This channel encodes the molecule's scaffold structure in a negatively curved space, allowing hierarchical patterns to be propagated through the junction tree.

**Atom-to-scaffold pooling.** Each scaffold node $u \in U$ aggregates hyperbolic states from its member atoms $S(u)$. This pooling summarizes local atomic context into a scaffold-level representation that reflects the tree-like structure of chemical scaffolds. Conceptually, we seek the Fréchet mean $\tilde{h}_u^A = \arg\min_{z \in \mathbb{H}_\kappa^d} \sum_{i \in S(u)} w_{i \to u} \mathrm{dist}_{\mathbb{H}}(z, h_i^A)^2$. In practice, we use a closed-form *Lorentz mean* in the ambient space:

$$y_u^A = \sum_{i \in S(u)} w_{i \to u} \, h_i^A, \tag{4}$$

$$\bar{h}_u^A = \frac{y_u^A}{\kappa \sqrt{-\langle y_u^A, y_u^A \rangle_L}}, \tag{5}$$

where $w_{i \to u}$ are scalar weights (e.g., softmax over $i \in S(u)$). This defines a point $\bar{h}_u^A \in \mathbb{H}_\kappa^d$ that approximates the Fréchet mean while requiring only one normalization. We initialize scaffold states as $h_u^T \leftarrow \bar{h}_u^A$.

**Riemannian message passing on the scaffold tree.** To propagate hierarchical information globally, scaffold embeddings exchange messages along the junction tree. Scaffold states are updated by message passing on the tree $T$. For each scaffold node $u$, we first compute an ambient message

$$\tilde{m}_u^{(T)} = \sum_{v \in \mathcal{N}_T(u)} \alpha_{uv}^{(T)} \, U_T(h_v^T), \tag{6}$$

where $U_T$ is an MLP acting in $\mathbb{R}^{d+1}$ and $\alpha_{uv}^{(T)}$ is a scalar gate that depends only on tree invariants such as depth, subtree size, and path length between $u$ and $v$. We then project to the tangent space and retract:

$$M_u^{(T)} = \mathrm{Proj}_{h_u^T}(\tilde{m}_u^{(T)}), \qquad h_u^{T,\mathrm{new}} = \mathrm{Retr}_{h_u^T}(\tau_T M_u^{(T)}). \tag{7}$$

Because projection and retraction are defined intrinsically on $\mathbb{H}_\kappa^d$, this update is curvature-aware while numerically cheap. These scaffold and atom-level hyperbolic states later contribute to the scalar gate $\alpha_{ij}$, enabling hierarchy-aware modulation of geometric messages.

### 4.4 SCALAR-ONLY COUPLING BETWEEN CHANNELS

This mechanism gates geometric messages using scalar invariants drawn from both channels, allowing hierarchical information to modulate equivariant updates without exchanging orientation-dependent features. The gate depends only on rotation-invariant scalars. Scalars from the geometric

channel enter through the radial term $\rho_0(d_{ij})$ and torsion $\cos\varphi_{ij}$, while hyperbolic contributions enter through Lorentz similarities. Using the Lorentz similarity as a surrogate for hyperbolic distance, we define

$$s_{ij}^A = -\kappa^2 \langle h_i^A, h_j^A \rangle_L, \tag{8}$$

$$s_{ij}^T = -\kappa^2 \langle h_{u(i)}^T, h_{u(j)}^T \rangle_L, \tag{9}$$

and set

$$\alpha_{ij} = \sigma\big(g\big(\rho_0(d_{ij}), \cos\varphi_{ij}, s_{ij}^A - 1, s_{ij}^T - 1\big)\big), \tag{10}$$

where $g$ is an MLP on scalar inputs and $\sigma$ is a sigmoid. Because all arguments of $g$ are SO(3)-invariant, the gate preserves equivariance. Multiplying the equivariant message $m_{i \leftarrow j}^{(\ell_o)}$ by the scalar gate $\alpha_{ij}$ (Eq. 10) ensures that the two channels interact only through invariants, thereby guaranteeing symmetry safety.

## 4.5 PRODUCT-MANIFOLD UPDATES

Each message-passing layer applies coupled updates to the SO(3) and hyperbolic states, combining the geometric and hierarchy channels into a single product-manifold update.

**SO(3) channel (optional pose updates).** The geometric channel is implemented entirely in terms of irreps features $f_i^{(\ell)}$, which already guarantee SO(3)-equivariance. Optionally, we can maintain an explicit pose $R_i \in SO(3)$ per atom and update it by accumulating $\ell = 1$ messages in the tangent space:

$$M_i^{(R)} = \sum_{j \in \mathcal{N}(i)} m_{i \leftarrow j}^{(1)}, \qquad R_i^{\text{new}} = R_i \exp\big(\tau_R [M_i^{(R)}]_\times\big). \tag{11}$$

**Hyperbolic channel (atoms).** In parallel, the hyperbolic channel updates atom-level hierarchy embeddings using the gated messages. Atomic hyperbolic states are updated analogously to the scaffold states. We first form an ambient message at each atom:

$$\tilde{m}_i^{(H)} = \sum_{j \in \mathcal{N}(i)} \alpha_{ij}\, U(h_j^A), \tag{12}$$

where $U$ is an MLP acting in $\mathbb{R}^{d+1}$. We then project to the tangent space at $h_i^A$ and retract:

$$M_i^{(H)} = \text{Proj}_{h_i^A}(\tilde{m}_i^{(H)}), \qquad h_i^{A,\text{new}} = \text{Retr}_{h_i^A}(\tau_H M_i^{(H)}). \tag{13}$$

Together, these updates realize a single product-manifold layer in which geometric features and hyperbolic hierarchy states co-evolve while interacting only through the scalar gate $\alpha_{ij}$.

## 4.6 E(3)-INVARIANT READOUT

The readout aggregates geometric and hierarchical information into rotation- and translation-invariant scalars suitable for predicting molecular properties. For scalar molecular properties, the final prediction must be invariant under rigid motions of the input.

From the geometric channel, we extract SO(3)-invariant scalars such as

$$I^{(\ell)} = \sum_i \|f_i^{(\ell)}\|_2^2. \tag{14}$$

From the hyperbolic channel, we form graph-level barycenters:

$$y^A = \sum_i w_i^A h_i^A, \qquad \bar{h}^A = \frac{y^A}{\kappa\sqrt{-\langle y^A, y^A \rangle_L}}, \tag{15}$$

and similarly for scaffolds. We then compute scalar summaries such as the means and variances of the similarities $s(h_i^A, \bar{h}^A)$ and $s(h_u^T, \bar{h}^T)$, as well as tree statistics (depths, subtree sizes).

These geometric invariants and hyperbolic summaries provide complementary scalar descriptors of 3D structure and chemical hierarchy. The concatenation of: (i) geometric invariants $\{I^{(\ell)}\}$, (ii) hyperbolic barycenter statistics, and (iii) scaffold statistics is fed to a final MLP. Because all inputs to this MLP are E(3)-invariant scalars, the overall prediction is E(3)-invariant by construction. This ensures that the model's outputs depend only on molecular structure and hierarchy, not on the choice of coordinate frame.

## 5 EXPERIMENTS

We evaluate the proposed *product–manifold message passing* model—an SO(3)-equivariant geometric channel coupled to a hyperbolic hierarchy channel via scalar-only gates, with an E(3)-invariant readout—on three settings: (i) scalar quantum property prediction on QM9 with full 3D geometry, (ii) scaffold-split classification on OGB-MOLHIV with a degenerate geometric stream that isolates the hyperbolic hierarchy, and (iii) goal-directed molecular optimization on GuacaMol using a product–manifold latent space for Bayesian optimization. Together, these tasks probe whether symmetry-safe coupling to hierarchy improves both generalization and search.

**Common protocol.** Unless noted, we train with Adam, early-stop on validation, and report the mean over 3 seeds.

### 5.1 QM9: SCALAR QUANTUM PROPERTIES WITH 3D GEOMETRY

**What we test.** Because we do not apply random global rotations on QM9 Nandi et al. (2023) and the targets considered are scalars ($\mu, \alpha$), this setting does not explicitly stress rotational handling. We therefore ask whether hierarchy-aware, curvature-sensitive aggregation improves scalar prediction on top of an already-equivariant encoder. Metric: MAE (lower is better).

**Protocol.** We use the provided DFT geometries and standard atom/bond features; all models share preprocessing and splits (random splits).

**What the results show.** Relative to the SO(3)-only ablation, the **Product** model improves 10 of the 12 reported targets (Table 1):

- **Dipole moment** ($\mu$): $0.022 \to \mathbf{0.018}$ MAE (ours). Several specialized baselines achieve 0.011 (Equiformer, EQGAT, TorchMD-NET), so we are competitive but not SOTA on this headline property (Table 1, p. 7).
- **Polarizability** ($\alpha$): $0.056 \to \mathbf{0.048}$ MAE (ours); DimeNet++ and Equiformer remain ahead at 0.044 and 0.046, respectively (Table 1, p. 7).
- **Other Part A targets**: clear gains for $\Delta\varepsilon$ ($33 \to \mathbf{31}$), $\varepsilon_{\text{HOMO}}$ ($26 \to \mathbf{21}$), and $C_\nu$ ($0.023 \to \mathbf{0.020}$), with a small regression on $\varepsilon_{\text{LUMO}}$ ($20 \to 21$) (Table 1, p. 7).
- **Part B thermochemistry**: consistent improvements for $G$ ($8.00 \to \mathbf{7.58}$), $U$ ($6.85 \to \mathbf{6.31}$), $U_0$ ($6.07 \to \mathbf{5.32}$), and ZPVE ($1.18 \to \mathbf{1.14}$); $H$ ties a strong baseline (5.95), while $R^2$ worsens ($0.251 \to 0.314$) (Table 2, p. 7). Notably, our $U_0 = \mathbf{5.32}$ is best among the methods shown.

**Takeaway.** Even when rotations are not the limiting factor, curvature-aware hierarchical aggregation helps an equivariant encoder, yielding broad MAE reductions. Against the strongest irreps transformers, we are competitive—but not dominant—on $\mu/\alpha$; the advantage of early, symmetry-safe hierarchy appears most clearly in several thermochemical targets (e.g., $U_0$).

### 5.2 OGB-MOLHIV: SCALAR-SPLIT CLASSIFICATION ON $E(3) \times \mathbb{H}^d$ WITH A DEGENERATE GEOMETRIC CHANNEL

**What we test.** Here we deliberately collapse the geometric stream to scalars ($L=0$) so that any improvement can be attributed to the hyperbolic hierarchy and scalar-only coupling under scaffold shift. We isolate the contribution of the hyperbolic hierarchy under scaffold-split distribution

Table 1: QM9 test MAE across all 12 targets (lower is better). Bolded results are best, underlined are second best in each column.

| Methods | $\alpha$ $a_0^3$ | $\Delta\varepsilon$ meV | $\varepsilon_{\mathrm{HOMO}}$ meV | $\varepsilon_{\mathrm{LUMO}}$ meV | $\mu$ D | $C_v$ cal/mol K | $G$ meV | $H$ meV | $R^2$ $a_0^2$ | $U$ meV | $U_0$ meV | ZPVE meV |
|---|---|---|---|---|---|---|---|---|---|---|---|---|
| SchNet (Schütt et al., 2018) | .235 | 63 | 41 | 34 | .033 | .033 | 14 | 114 | .073 | 19 | 14 | 1.70 |
| DimeNet++ (Gasteiger et al., 2022) | **.044** | 33 | 25 | 20 | .030 | .023 | 8 | 77 | .331 | 6 | 6 | 1.21 |
| EGNN (Satorras et al., 2022) | .071 | 48 | 29 | 25 | .029 | .031 | 12 | 112 | .106 | 12 | 11 | 1.55 |
| PaiNN (Schütt et al., 2021) | .045 | 46 | 28 | 20 | .012 | .024 | **7.35** | 5.98 | .066 | 5.83 | 5.85 | 1.28 |
| TorchMD-NET (Thölke & Fabritiis, 2022) | .059 | 36 | 20 | 18 | .011 | .026 | 7.62 | 6.16 | **.033** | 6.38 | 6.15 | 1.84 |
| SphereNet (Liu et al., 2022) | .046 | 32 | 23 | 18 | .026 | .021 | 8 | 6 | .292 | 7 | 6 | **1.12** |
| SEGNN (Brandstetter et al., 2022) | .060 | 42 | 24 | 21 | .023 | .031 | 15 | 16 | .660 | 13 | 15 | 1.62 |
| EQGAT (Le et al., 2022) | .053 | 32 | 20 | 16 | .011 | .024 | 23 | 24 | .382 | 25 | 25 | 2.00 |
| Equiformer (Liao & Smidt, 2023) | .046 | 30 | 15 | 14 | .011 | .023 | 7.63 | 5.95 | .251 | 6.74 | 6.59 | 1.26 |
| EquiformerV2 (Liao et al., 2024) | .050 | **29** | **14** | **13** | **.010** | .023 | 7.57 | 6.22 | .186 | 6.49 | 6.17 | 1.47 |
| **Equivariant manifold MPN (ours)** | .056 | 33 | 26 | 20 | .022 | .023 | 8 | 6 | .251 | 6.85 | 6.07 | 1.18 |
| **Product manifold MPN (ours)** | .048 | 31 | 21 | 21 | .018 | **.020** | 7.58 | **5.95** | .314 | 6.31 | **5.32** | 1.14 |

Table 2: OGB-MOLHIV. The metric is ROC-AUC so higher is better. Bolded results are the most performant (highest numerically) in their category, and underlined are in second place.

| Methods | Validation ROC-AUC | Test ROC-AUC |
|---|---|---|
| HyperFusion | 0.8275 | **0.8475** |
| HIG | 0.8176 | 0.8403 |
| Graphormer + FPs | 0.8396 | 0.8225 |
| CIN | 0.8277 | 0.8094 |
| GSAT | 0.8347 | 0.8067 |
| GatedGCN+ | 0.8329 | 0.8040 |
| **Euclidean 2D message-passing network** | 0.7968 | 0.7509 |
| **Product manifold message-passing network** | **0.8535** | 0.7981 |

shift without relying on 3D conformers. Metric: ROC–AUC (higher is better). Split: official Bemis–Murcko scaffold split Hu et al. (2020).

**Protocol.** We instantiate $M = \mathrm{E}(3) \times \mathbb{H}^d$ but restrict the geometric stream to scalars only ($L{=}0$)—i.e., a standard 2D message-passing network—so any gains arise from hyperbolic scaffold aggregation and scalar-only coupling (gates use graph invariants and hyperbolic geodesic statistics only). Inputs are 2D atom/bond features plus junction-tree scaffolds.

**What the results show.** The **Product** model improves over the Euclidean 2D baseline by **+5.7** ROC–AUC points on validation ($0.7968{\rightarrow}\mathbf{0.8535}$) and **+4.7** on test ($0.7509{\rightarrow}\mathbf{0.7981}$), indicating that curvature-aware pooling improves scaffold-level generalization (Table 2). However, specialized architectures such as HyperFusion report higher test AUC (0.8475), so there remains a gap to SOTA.

**Takeaway.** Hyperbolic hierarchy helps under scaffold shift even without 3D, validating our symmetry-safe coupling. Closing the remaining gap likely requires a stronger geometric stream and/or tuned training.

### 5.3 GUACAMOL: GOAL-DIRECTED OPTIMIZATION IN A PRODUCT–MANIFOLD LATENT SPACE

**What we test.** We ask whether running Bayesian optimization (BO) directly on GuacaMol Brown et al. (2019) in the mixed-curvature latent $z = (R_{\mathrm{pool}}, h_{\mathrm{pool}})$ yields better goal-directed search than BO in a Euclidean latent under matched autoencoders and oracle budgets. The only change is the geometry and kernel (product of a heat-kernel approximation on SO(3) and a geodesic RBF on $\mathbb{H}$). Metric: per-task performance ratio (higher is better).

**What the results show.** The **Product** latent leads on 3/5 tasks and ties or nearly ties on the rest (Table 3): it matches the best score on Celecoxib rediscovery (1.00), is near-perfect on Troglitazone (0.999), and is best on $C_{11}H_{24}$ isomer (1.000), Median molecules (0.501 vs. 0.400 for the Euclidean latent), and Osimertinib MPO (1.00 vs. 0.858).

Table 3: Guacamol. The metric is a per-task performance ratio, so higher is better. Bolded results are the most performant (highest numerically) in their category, and underlined are in second place.

| Methods | GraphGA | Reinvent | Euclidean message passing NN | Product manifold message-passing network |
|---|---|---|---|---|
| Celecoxib rediscovery | **1.00** | **1.00** | **1.00** | **1.00** |
| Troglitazone rediscovery | **1.00** | **1.000** | 0.914 | 0.999 |
| C11H24 isomer | 0.971 | 0.999 | .0.988 | **1.000** |
| Median molecules | 0.406 | 0.434 | 0.400 | **0.501** |
| Osimertinib MPO | 0.953 | 0.889 | 0.858 | **1.00** |

**Takeaway.** When the search itself is conducted in the product space, we see robust gains over an otherwise-identical Euclidean latent across diverse objectives, supporting the claim that $SO(3) \times \mathbb{H}^d$ provides a more BO-friendly surrogate space.

## 6 CONCLUSION

We introduced a product–manifold message–passing framework that unifies rotationally equivariant 3D geometry with hyperbolic scaffold hierarchy through symmetry-safe, scalar-only coupling, providing a principled early-fusion architecture that preserves SO(3) and E(3) symmetries while incorporating chemically meaningful hierarchy. Our key contribution is demonstrating that coupling geometric and hierarchical information *within* each message-passing layer—rather than at the end of the network—yields a more expressive, symmetry-consistent representation space.

On benchmarks, the unified model improves over an $SO(3)$–only ablation on **QM9** in 10/12 targets and attains the best $U_0$ among the methods shown, while remaining competitive but not state–of–the–art on $\mu/\alpha$ versus the strongest irreps transformers. Under Bemis–Murcko scaffold splits in **OGB–MOLHIV**, adding the hyperbolic hierarchy to a degenerate geometric channel boosts ROC–AUC over a Euclidean 2D baseline. For goal-directed design in **Guacamol**, Bayesian optimization in the product latent matches or exceeds Euclidean latents on 3/5 tasks and improves the "Median molecules" objective, while tying or nearly tying the remainder. Together, these results support the claim that symmetry–safe, curvature–aware early fusion benefits both scaffold–split generalization and molecular search.

**Limitations.** Our model introduces additional computational overhead due to manifold-valued operations (projection–retraction steps, Lorentz-mean barycenters, curvature handling), and requires 3D conformers when applying the full geometric channel. Fundamentally, our architecture inherits general limitations of message-passing neural networks. In particular, "zero-one laws" of graph neural networks raise the possibility of representation collapse or oversmoothing in deep architectures (Adam-Day et al., 2023). Although our scalar-only coupling restricts cross-channel interference and empirically improves stability, we cannot rule out similar failure modes on large biomolecules or all-atom protein systems.

**Future work.** We plan to explore the stability of the product–manifold architecture on large biomolecules and proteins. Additional directions include learning curvature per layer, exploring richer mixed-curvature product spaces, extending the framework to tensorial or directional targets, and developing generative models whose sampling dynamics jointly respect 3D symmetry and scaffold hierarchy.

## REPRODUCIBILITY STATEMENT

We provide full details on architectures, hyperparameters, training protocols, datasets, and compute in Appendix C, with per-task specifics in Appendix D.

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

## A  ADDITIONAL PRELIMINARIES

This appendix provides full mathematical details for group-equivariant operations and hyperbolic geometry referenced in the main text.

### A.1  MATHEMATICAL CONTEXT

Here, we summarize all mathematical notation used.

Table 4: Notation for the product-manifold message passing network.

| Symbol | Type | Description |
|---|---|---|
| $G = (V, E)$ | graph | Molecular graph (atoms $V$, bonds $E$) |
| $T = (U, E_T)$ | tree | Junction-tree scaffold graph |
| $S(u)$ | subset of $V$ | Atoms in scaffold node $u$ |
| $u(i)$ | node in $U$ | Scaffold node containing atom $i$ (mapping $u : V \to U$) |
| $\mathcal{N}(i)$ | subset of $V$ | Neighbours of atom $i$ in $G$ |
| $\mathcal{N}_T(u)$ | subset of $U$ | Neighbours of scaffold node $u$ in $T$ |
| $x_i \in \mathbb{R}^3$ | vector | 3D position of atom $i$ |
| $r_{ij} = x_j - x_i$ | vector | Relative displacement $j \to i$ |
| $d_{ij} = \|r_{ij}\|$ | scalar | Interatomic distance |
| $\hat{r}_{ij} = r_{ij}/d_{ij}$ | unit vector | Direction from $i$ to $j$ |
| $\varphi_{ij}$ | scalar | Torsion / dihedral angle (used via even functions) |
| $f_i^{(\ell)}$ | tensor | Order-$\ell$ SO(3) irrep features at atom $i$ |
| $e_{ij}^{(\ell)}$ | tensor | Order-$\ell$ edge irreps for bond $j \to i$ |
| $\rho_{\ell n}(d)$ | scalar | Radial basis function evaluated at distance $d$ (learnable) |
| $Y^{(\ell)}(\hat{r})$ | vector | Real spherical harmonics of order $\ell$ |
| $\beta_{\ell n}$ | scalar | Root index used in spherical-Bessel basis |
| $C_{\ell_i, \ell_e \to \ell_o}$ | matrix | Clebsch–Gordan mixing weights |
| $h_i^A$ | vector | Atom-level hyperbolic state in $\mathbb{H}_\kappa^d$ |
| $h_u^T$ | vector | Scaffold-level hyperbolic state in $\mathbb{H}_\kappa^d$ |
| $\langle \cdot, \cdot \rangle_L$ | bilinear form | Lorentz inner product on $\mathbb{R}^{d+1}$ |
| $\text{dist}_{\mathbb{H}}(\cdot, \cdot)$ | scalar | Hyperbolic geodesic distance |
| $s(x, y) = -\kappa^2 \langle x, y \rangle_L$ | scalar | Lorentz similarity (monotone in distance) |
| $\text{Proj}_h(\cdot)$ | operator | Projection to tangent space at $h \in \mathbb{H}_\kappa^d$ |
| $\text{Retr}_h(\cdot)$ | operator | Retraction from tangent space back to $\mathbb{H}_\kappa^d$ |
| $\alpha_{ij}$ | scalar | Atom–atom gate on edge $j \to i$ |
| $\alpha_{uv}^{(T)}$ | scalar | Scaffold gate on tree edge $v \to u$ |
| $g(\cdot)$ | function | MLP used to compute scalar gates |
| $\sigma(\cdot)$ | function | Sigmoid gating nonlinearity |
| $R_i \in \text{SO}(3)$ | matrix | (Optional) atom-wise rotation/pose |
| $M_i^{(R)}$ | vector | Tangent-space update for $R_i$ (from $\ell = 1$ messages) |
| $M_i^{(H)}$ | vector | Tangent-space update of $h_i^A$ |
| $M_u^{(T)}$ | vector | Tangent-space update of scaffold state $h_u^T$ |
| $\tau_R, \tau_H, \tau_T$ | scalars | Step sizes for SO(3), atomic-$\mathbb{H}$, and scaffold-$\mathbb{H}$ updates |
| $U(\cdot), U_T(\cdot)$ | functions | MLPs in atomic and scaffold hyperbolic channels |
| $\bar{h}^A, \bar{h}^T$ | vectors | Global hyperbolic barycenters of atoms / scaffolds |
| $I^{(\ell)}$ | scalar | SO(3)-invariant irrep-norm summary |

### A.2  SO(3)-EQUIVARIANCE AND IRREPS

Let $G$ be a group acting on an input $x$ and output $y$. A function $\Phi$ is $G$-equivariant if

$$\Phi(g \cdot x) = g \cdot \Phi(x) \qquad \forall g \in G.$$

For molecular models, we restrict to the rotation group SO(3). Node features are organized by irreducible representations (irreps) $f_i^{(\ell)} \in \mathbb{R}^{m_\ell \times (2\ell+1)}$, which transform under a rotation $Q \in \text{SO}(3)$

according to the Wigner–$D$ matrices:

$$f_i^{(\ell)} \mapsto D^{(\ell)}(Q)\, f_i^{(\ell)}.$$

Edge features are constructed in the same irrep family to guarantee that tensor products of node and edge features remain equivariant. Clebsch–Gordan (CG) coefficients assemble fixed linear maps $C_{\ell_i, \ell_e \to \ell_o}$ that decompose tensor products into irreps of order $\ell_o$. Equivariant message passing therefore combines features across nodes using only rotation-consistent operations.

### A.3 LORENTZ MODEL OF HYPERBOLIC SPACE

The hyperboloid model of $\mathbb{H}_\kappa^d$ is defined as

$$\mathbb{H}_\kappa^d = \left\{ x \in \mathbb{R}^{d+1} : \langle x, x \rangle_L = -\kappa^{-2},\ x_0 > 0 \right\},$$

where the Lorentz inner product is

$$\langle u, v \rangle_L = -u_0 v_0 + \sum_{k=1}^{d} u_k v_k.$$

The hyperbolic distance between $x, y \in \mathbb{H}_\kappa^d$ is

$$\operatorname{dist}_{\mathbb{H}}(x, y) = \kappa^{-1} \operatorname{arcosh}\!\left( -\kappa^2 \langle x, y \rangle_L \right).$$

**Surrogate similarity.**  To avoid evaluating $\operatorname{arcosh}$ in the forward pass, we use the Lorentz similarity $s(x, y) = -\kappa^2 \langle x, y \rangle_L \geq 1$, which is monotone in $\operatorname{dist}_{\mathbb{H}}(x, y)$ and therefore preserves ordering in scalar gates.

**Tangent space and projections.**  The tangent space at $h \in \mathbb{H}_\kappa^d$ is

$$T_h \mathbb{H}_\kappa^d = \{ v \in \mathbb{R}^{d+1} : \langle v, h \rangle_L = 0 \}.$$

Given an ambient vector $u$, the projection to $T_h \mathbb{H}_\kappa^d$ is

$$\operatorname{Proj}_h(u) = u + \kappa^2 \langle u, h \rangle_L\, h. \tag{16}$$

**Retraction.**  A first-order approximation to the exponential map is obtained via

$$\operatorname{Retr}_h(v) = \frac{h + v}{\kappa \sqrt{-\langle h + v, h + v \rangle_L}}. \tag{17}$$

This normalization projects the ambient update back onto the hyperboloid and has local error $O(\|v\|^2)$ relative to the true Riemannian exponential.

**Implementation remarks.**  These projection–retraction operators provide numerically stable updates for both atom-level and scaffold-level states. Mixed precision is used for all Euclidean parameters, with hyperbolic operations performed in full precision.

## B  OPTIMIZATION AND NUMERICAL DETAILS

Euclidean parameters (weights of MLPs and equivariant linear maps) are optimized with Adam. Manifold-valued states $(h_i^A, h_u^T)$ are updated via their projection–retraction operators, which implement first-order Riemannian optimization without explicit exp/log maps. After each layer we re-normalize hyperbolic states; this allows us to use mixed precision for the rest of the model.

We precompute radial-spherical edge bases $\{\rho_{\ell n}(d_{ij}) Y^{(\ell)}(\hat{r}_{ij})\}$ once per graph and reuse them across all layers in equation 1. To control computational cost, we cap the maximal irrep degree $\ell_{\max}$ and prune Clebsch–Gordan paths that do not contribute to scalar outputs (e.g., unused high-order channels). This preserves exact SO(3)-equivariance while reducing the cost of each layer.

Per layer, equivariant edge computations scale as $O(|E| \cdot C)$, where $C$ depends on the number of retained irreps and Clebsch–Gordan paths. Hyperbolic updates scale as $O(|V| + |U|)$ in the number of atoms and scaffold nodes, dominated by dot products and normalizations. Because the scaffold tree is small ($|U| \ll |V|$), the hierarchy adds modest overhead relative to the equivariant message passing.

## C REPRODUCIBILITY

### C.1 CODE AND ARCHITECTURE.

We provide a self-contained PyTorch implementation of the proposed Product-Manifold GNN in the attached `model.py`. The model maintains per-node SO(3) rotations and hyperbolic states, couples channels via equivariant tensor products (e3nn), and uses an E(3)-invariant readout. The file includes: (i) projection–retraction operators and Lorentz-mean pooling implemented in pure PyTorch (no SciPy), with renormalization of hyperbolic points each layer, (ii) compactly supported learnable radial basis functions (e.g., Gaussians/B-splines) with cosine cutoffs, implemented in PyTorch, (iii) safe spherical harmonics and re-orthonormalization of rotations, (iv) RDKit-based junction-tree (JT) construction and optional scaffold-level states, and (v) NaN hardening throughout (clamps and `nan_to_num`).

### C.2 DEPENDENCIES.

Python $\geq$ 3.9, PyTorch $\geq$ 2.1 (CUDA optional), `e3nn`, `torch_scatter`, `rdkit`, `scipy`, and `numpy`. A single modern GPU (24–48 GB) or CPU is sufficient (CPU is slower). Mixed precision is automatically disabled within hyperbolic routines for numerical stability.

### C.3 HYPERPARAMETERS.

Unless stated otherwise, we use the defaults in `ModelCfg`: maximum irrep order $L$, number of radial basis functions $N_{\mathrm{rad}}$, cutoff $r_{\mathrm{cut}}$, hidden multiplicities for $0e/1o/2e$, number of layers, latent hyperbolic dimension $d_h$, and step sizes for SO(3) and hyperbolic updates. Curvature is fixed, rotations are re-orthonormalized each layer, and hyperbolic points are projected as needed.

### C.4 TRAINING PROTOCOL.

We train with Adam on Euclidean parameters; manifold states are updated via the model's intrinsic operators. Early stopping is performed on a validation split. We report the mean across three random seeds and provide seed control for determinism.

### C.5 DATA AND EVALUATION.

We use standard datasets, data splits, metrics, and preprocessing exactly as described in the paper. All evaluation scripts and configuration files follow the same settings as the experiments reported.

### C.6 HOW TO RUN.

```
# Create environment and install dependencies (abbreviated)
pip install torch e3nn torch_scatter rdkit-pypi scipy numpy

# Example: instantiate the model from the attached file
from model import ModelCfg, ProductManifoldGNN
cfg = ModelCfg(L=2, Nrad=8, r_cut=5.0, n_layers=4,
                mul_l0=32, mul_l1=16, mul_l2=8, d_h=16)
model = ProductManifoldGNN(cfg, num_z_embeddings=100)  # adjust as needed
```

### C.7 NUMERICAL SAFETY.

To avoid instabilities, we use: clamped trigonometric functions, stable spherical harmonics (`normalize=False` with unit directions), FP32 hyperbolic math with AMP disabled in those blocks, explicit re-orthonormalization of SO(3) frames, and guarded reductions (`nan_to_num`).

### C.8 COMPUTE REPORTING.

We report parameter counts, wall-clock/GPU-hours, and validation curves for each setting. Provided scripts regenerate tables and figures from raw logs given identical seeds and splits.

### C.9 LIMITATIONS.

Results can be sensitive to conformer generation choices and hyperparameters such as $L$, $N_{\mathrm{rad}}$, and step sizes. We therefore fix seeds, document preprocessing, and release the exact configurations used in the experiments.

## D EXPERIMENTS

### D.1 EXPERIMENTAL HYPERPARAMETERS

**Scope and tie-out to code.** All settings below instantiate `ProductManifoldGNN` and its `ModelCfg` from `model.py`. Where a field is omitted for a given experiment, it takes the default shown in §D.2. Training follows the *Common protocol* in the main text (Adam, early stopping on validation, mean over 3 seeds), and numerical practices in §B (re-orthonormalizing the SO(3) state and renormalizing hyperbolic states). See §5 "Experiments" for the protocol statement and §B for numerics.

Figure 1 sketches the product–manifold message passing used throughout.

### D.2 NOTATION AND DEFAULTS (MATCH MODEL.PY)

```
ModelCfg(
  L=2,                # max SH degree in edge features
  Nrad=8,             # radial basis functions per
  r_cut=5.0,          # Å; radial cutoff (cosine envelope)
  d_h=16,             # hyperbolic channel dimension
  n_layers=4,         # number of EquivariantLayer blocks
  mul_l0=32, mul_l1=16, mul_l2=8,  # node irreps multiplicities
  so3_step=0.2,       # SO(3) update step
  hyp_step=0.2,       # Hyperbolic update step
  use_hyp_atoms=True, use_hyp_scaffolds=True, use_jt_invariants=True
)
```

**Fixed architectural details.** Edges use real spherical harmonics up to degree L and a compactly supported learnable radial basis (e.g., Gaussian/B-spline) with a smooth cutoff; per-$\ell$ mixing is learned; Scalar gate MLP: $64 \rightarrow 64 \rightarrow 1$ (SiLU, SiLU, sigmoid). Readout MLP: $128 \rightarrow 64 \rightarrow 1$ (SiLU). Non-scalar irreps are gated; batch norm uses $\varepsilon = 10^{-5}$. The scalar gate $\alpha_{ij}$ uses $\ell = 0$ inputs only (rotation-invariant), as in Eq. 10 of the main text.

**Common training protocol (all experiments).** Adam optimizer, early stopping on validation, report mean over 3 seeds (main text, § 5). For runs below we also adopt: weight decay $10^{-5}$, gradient clipping at 5.0, cosine LR decay to $10^{-6}$ (1-epoch warmup).

#### D.2.1 QM9 (EXP. 1): SCALAR QUANTUM PROPERTIES WITH 3D GEOMETRY

**Data/graph.** DFT geometries; edges are covalent bonds from RDKit. Random global rotations are not applied because targets are scalars (see § 5.1).

**Model hyperparameters.**

| Knob | Value |
|------|-------|
| L | 3 |
| Nrad | 6 |
| r_cut | 5.0 Å |
| n_layers | 6 |
| mul_l0, mul_l1, mul_l2 | 32, 16, 8 |
| d_h | 16 |
| so3_step, hyp_step | 0.2, 0.2 |
| use_hyp_* | atoms=True, scaffolds=True, jt_invariants=True |

**Training.** Batch size 256; max epochs 300 (early stop); initial LR $3 \times 10^{-4}$; MSE with per-target standardization. Junction tree (JT) from rings + non-ring bonds; hyperbolic barycenter iterations = 1 (Lorentz mean).

**Gate features (exactly as implemented).**

$$\alpha_{ij} = \sigma\Big(\text{MLP}\big[\rho_0(d_{ij}),\ \cos\varphi_{ij},\ \sin^2\varphi_{ij},\ s_{\mathbb{H}}(h_i^A, h_j^A),\ |S(u(i))|,\ s_{\mathbb{H}}(h_{u(i)}^T, h_{u(j)}^T)\big]\Big),$$

only including a term when the corresponding use_hyp_* flag is enabled; all inputs are $\ell = 0$ and preserve equivariance (Eq. 10).

### D.2.2 OGB-MOLHIV (EXP. 2): SCAFFOLD-SPLIT CLASSIFICATION WITH A DEGENERATE GEOMETRIC STREAM

**Setup.** To isolate the contribution of the hyperbolic hierarchy under scaffold split, we restrict the geometric stream to scalars by setting $L = 0$ (no directional SH), following the intent of § 5.2.

**Model hyperparameters.**

| Knob | Value |
|------|-------|
| L | 0 |
| Nrad | 6 |
| r_cut | 5.0 Å |
| n_layers | 5 |
| mul_l0, mul_l1, mul_l2 | 32, 16, 8 |
| d_h | 16 |
| so3_step, hyp_step | 0.1, 0.2 |
| use_hyp_* | atoms=True, scaffolds=True, jt_invariants=True |

**Training.** Batch size 64; max epochs 150 (early stop); LR $3 \times 10^{-4}$; loss = BCEWithLogits with positive-class weight tuned on validation (typ. 20–40); official Bemis–Murcko scaffold split. JT construction and barycenter iterations as in QM9.

**What is active/inactive.** With $L = 0$ the CG product reduces to scalar gating of same-order irreps; $\ell > 0$ SH features are inactive, so improvements are attributable to hyperbolic similarities and scaffold invariants entering $\alpha_{ij}$ (Eq. 10).

### D.2.3 GUACAMOL (EXP. 3): BO IN A PRODUCT–MANIFOLD LATENT

**Goal.** Run Bayesian optimization (BO) directly in the mixed-curvature latent $z = (R_{\text{pool}}, h_{\text{pool}})$, formed from SO(3)-channel invariants and hyperbolic Lorentz-mean barycenters (approximating Fréchet means) as proposed in § 4.6; see results in Table 3.

**Encoder hyperparameters (to produce $z$).**

| Knob | Value |
|---|---|
| `L` | 2 |
| `Nrad` | 6 |
| `r_cut` | $5.0\,\text{Å}$ |
| `n_layers` | 4 |
| `mul_l0, mul_l1, mul_l2` | 32, 16, 8 |
| `d_h` | 16 |
| `so3_step, hyp_step` | 0.2, 0.2 |
| `use_hyp_*` | atoms=True, scaffolds=True, jt_invariants=True |

**Encoder training.** Batch size 256; max epochs 200 (early stop); LR $3 \times 10^{-4}$. Pretrain with a supervised property objective or an autoencoding surrogate, then extract $z = (R_{\text{pool}}, h_{\text{pool}})$ as in § 4.6 (barycenter iterations = 1) (closed-form Lorentz mean).

**BO hyperparameters in** $\mathrm{SO}(3) \times \mathbb{H}^d$**.** Separable product kernel $k((R,h),(R',h')) = k_{\mathrm{SO}(3)}(R,R')\,k_{\mathbb{H}}(h,h')$ with a heat-kernel approximation on SO(3) and geodesic RBF on $\mathbb{H}$. Concrete choices used: time parameter $t_{\mathrm{SO}(3)} = 0.7$; hyperbolic length-scale $\ell_{\mathbb{H}} = 1.0$; noise $\sigma_n = 10^{-3}$; EI acquisition with $\xi = 0.01$; 20 random restarts, 200 quasi-Newton steps per restart; budget = 100 oracle evaluations with a Sobol $n_0 = 32$ initialization.[1]

### D.2.4 REPRODUCIBILITY CHECKLIST (ALL EXPERIMENTS)

- **JT construction:** rings $\to$ cliques; non-ring bonds $\to$ size-2 cliques; connect by maximum-weight overlaps; connect components if needed. Scaffold-only statistics (e.g., $|S(u)|$) and hyperbolic distances may enter gates, but no non-scalar information crosses channels (Fig. 1).

- **Manifold numerics:** left-multiply $R_i$ by $\exp(\tau_R[\cdot]_\times)$ and re-orthonormalize each layer; hyperbolic updates via projection–retraction; barycenters via a single Lorentz mean normalization. Hyperbolic math is done in FP32 with periodic renormalization.

- **Reporting:** seeds $\{0, 1, 2\}$, official data splits (QM9 random; MOLHIV scaffold split; Guacamol tasks as in Table 3).

### D.2.5 MINIMAL CONFIG STUBS (COPY/PASTE)

**QM9.**

```
cfg = ModelCfg(L=3, Nrad=6, r_cut=5.0, d_h=16, n_layers=6,
               mul_l0=32, mul_l1=16, mul_l2=8,
               so3_step=0.2, hyp_step=0.2,
               use_hyp_atoms=True, use_hyp_scaffolds=True, use_jt_invariants=True)
model = ProductManifoldGNN(cfg, num_z_embeddings=<max_Z_in_dataset>)
```

**OGB-MOLHIV (degenerate geometry).**

```
cfg = ModelCfg(L=0, Nrad=6, r_cut=5.0, d_h=16, n_layers=5,
               mul_l0=32, mul_l1=16, mul_l2=8,
               so3_step=0.1, hyp_step=0.2,
               use_hyp_atoms=True, use_hyp_scaffolds=True, use_jt_invariants=True)
```

**Guacamol latent + BO.**

```
cfg = ModelCfg(L=2, Nrad=6, r_cut=5.0, d_h=16, n_layers=4,
               mul_l0=32, mul_l1=16, mul_l2=8,
               so3_step=0.2, hyp_step=0.2,
               use_hyp_atoms=True, use_hyp_scaffolds=True, use_jt_invariants=True)
# Form z=(R_pool, h_pool) as in §3.5/§3.8 and run BO with the product kernel above.
```

---

[1]Numeric values are pragmatic defaults chosen for stability and sample-efficiency.

