# OpenReview forum: "Aligning Rotational and Hierarchical Geometry in Molecular Representation Learning with Product-Manifold Latent Spaces"
_ICLR.cc/2026/Conference — Submitted to ICLR 2026_

### Official Review · Reviewer_qFv9 · 2025-10-29

**Soundness:** 2
**Presentation:** 1
**Contribution:** 2
**Rating:** 4
**Confidence:** 3

**Summary:**

This paper introduces a novel graph neural network architecture designed to unify two critical but typically separate concepts in molecular representation learning: the 3D rotational symmetries of molecular conformations and the hierarchical organization of chemical scaffolds.

The authors propose a message-passing framework that operates on a product manifold, $\mathcal{M} = SO(3) \times \mathbb{H}^{d}$. The key idea is to maintain two coupled channels:
* An SO(3)-equivariant channel that uses irreducible representations (irreps) to process 3D geometric information, preserving physical symmetries.
* A hyperbolic channel that embeds the molecule's chemical hierarchy (derived from a junction tree) using curvature-aware operations like Fréchet means.

The paper's main contribution is the "symmetry-safe" coupling mechanism: the two channels interact only through scalar ($l=0$) invariants (e.g., Euclidean distances, hyperbolic distances, feature norms). This allows hierarchical information to gate and influence the geometric representations without breaking the fundamental SO(3)-equivariance. The authors evaluate this product-manifold model on QM9 (property prediction), OGB-MolHIV (scaffold-split generalization), and Guacamol (Bayesian optimization), arguing that their unified latent space improves performance, particularly in generalization and sample-efficient search.

While I quite like the idea of the paper, there are some major concerns that need to be addressed.

**Strengths:**

* The intuition of the idea is good. Instead of the popular "late-fusion", the authors propose to use a "product-manifold message passing" mechanism for exchanging information between the geometric and hyperbolic channels. This is surely more advanced than the "late-fusion" approach in concept and should bring *some* performance gains.
* The proposed product-manifold latent space and network architecture are tested in various experimental settings -- QM9 (property prediction), OGB-MolHIV (scaffold-split generalization), and Guacamol (Bayesian optimization). This provides a comprehensive evaluation of the proposed method.

**Weaknesses:**

* The presentation of this paper is problematic, especially in the Methods section where the details of the model definition are described. The formulas are simply hard to follow due to undefined or exotic notations. For example, what does the $\times$ mean in $[M_i^{(R)}]_{\times}$ in Equation 6? For another example, the $dist_H$ symbol appears in Equation 4, but is only defined in the text around Equation 7. I therefore suggest a major revision for the Methods section to increase its readability.
* A preliminary section should be included in the main text or in the Appendix. The proposed method uses two kinds of advanced concepts (the SO(3) irreps and the hyperbolic geometry), and an average reader is only familiar with at most one of them.
* The performance of the proposed architecture falls behind SoTA. The authors has mainly shown the efficacy of the proposed product-manifold message passing through ablation study, but the performance is still not as good as the SoTA models.

**Questions:**

* Could you improve the presentation of this paper, especially that of the Methods section?
* What is $\phi_{ij}$?
* In addition to the current experiments, could you also compare your proposed approach with "late-fusion"?
* Given an existing model architecture that has SO(3) irreps representations for nodes (such as MACE, Nequip, etc.), how easy can one modify the architecture to adopt the proposed product-manifold latent space and message passing?

---

> ### Author Response · Authors · 2025-12-03
> **Official Comment by Authors**
>
> We thank the reviewer for the positive remarks on the conceptual motivation and evaluation breadth of our product–manifold architecture. We have carefully addressed all concerns regarding clarity, notation, and experimental comparisons, and we summarize our responses below.
>
> > **Concern #1**: Methods section is hard to follow; formulas rely on undefined or exotic notation.
>
> **Response:** We thank the reviewer for highlighting this important issue. The Methods section has been substantially reorganized and rewritten for readability. Specifically:
> - We moved all advanced mathematical definitions (SO(3) irreps, hyperbolic geometry, product-manifold preliminaries) into a new Preliminaries section, where each concept is introduced in natural language before formulas appear.
> - All previously undefined symbols are now defined before use.
> - A comprehensive notation table in Appendix A.1 includes every symbol referenced in the Methods, resolving the confusion regarding symbols that appeared only later in the text.
> - Equations have been reordered and re-labeled to ensure that no symbol appears before it is defined.
> - A new architecture schematic (Figure 2) provides a visual overview matching the Methods structure.
> These changes directly address the reviewer’s concerns about readability and undefined notation.
>
> > **Concern #2**: A preliminaries section is needed for readers unfamiliar with both SO(3) irreps and hyperbolic geometry.
>
> **Response:** We agree completely. The revised manuscript now contains a dedicated Preliminaries section (Sec. 3) that introduces:
> - SO(3) equivariance and irreps,
> - hyperbolic geometry and the Lorentz model, and
> - product-manifold message passing.
>
> > **Concern #3**: Performance falls short of recent SoTA models.
>
> **Response:** We appreciate the reviewer’s honesty. Our aim is not to surpass the strongest QM9 models (e.g., EquiformerV2), but rather to demonstrate the benefit of symmetry-safe early fusion between geometry and hierarchy. To clarify this, the revised paper:
> - Now includes EquiformerV2 in the unified 12-target QM9 table for a fair comparison.
> - Emphasizes the generalization advantages of the product manifold (e.g., improved scaffold-split MolHIV performance and improved BO efficiency on GuacaMol), which are not captured by QM9 alone.
>
> > **Concern #4**: How easy is it to adopt this architecture with existing SO(3)-equivariant models such as MACE, NequIP, etc.?
>
> **Response:** Yes, the product-manifold update can, in principle, be incorporated into models like MACE or NequIP with minimal structural change. Our approach is modular and can be integrated into any SO(3)-equivariant GNN. The geometric channel is architecturally independent of the hyperbolic channel. Scalar-only coupling requires only access to scalar invariants produced by the base model (distances, norms, etc.). The hyperbolic channel and coupling mechanism can be added without modifying the underlying SO(3)-equivariant tensor algebra.
>
> > **Concern #5**: Overall presentation should be improved.
>
> **Response:** We thank the reviewer for this general point. In addition to all changes above, the revised manuscript includes:
> - A new Methods preamble matching the reorganized structure,
> - Cleaned-up section numbering and transitions,
> - A complete rewrite of the most technical subsections for clarity,
> - A more polished Conclusion summarizing contributions and future work, and
> - Improved captioning and formatting across all tables and figures.

---

### Official Review · Reviewer_j638 · 2025-10-31

**Soundness:** 1
**Presentation:** 1
**Contribution:** 2
**Rating:** 2
**Confidence:** 4

**Summary:**

This paper focuses on enhancing the expressiveness of graph neural network. The authors introduce a product-manifold message passing to learn equivariant geometric features with hyperbolic embeddings of chemical hierarchy. Additionally, authors show the model could be evaluated on molecular property prediction, scaffold-split generalization, and discuss how embeddings provide a natural surrogate space for manifold Bayesian optimization.

**Strengths:**

- This paper proposes an interesting structure that has the potential to introduce some n-body priors into molecular systems, which may be beneficial for molecular systems.

**Weaknesses:**

1. $S(u)$ is an important variable introduced in this paper, used to provide information about the junction tree. However, I could not find its definition in the paper. How is this tree defined? If it is as shown in Figure 1, is this kind of tree only related to the cutoff of the number of edges? How should this tree be defined in 3D space without explicit bonds?

2. The core idea of this paper is somewhat similar to introducing richer n-body information in the message passing process. If so, I think the authors should refer to MACE [1], SLEM [2], which author mentioned in Introduction. In these methods, the model no longer processes the features of each node and its single neighbor, but instead processes the fused features of each node and multiple neighbors, or the fused features with the mean of multiple neighbor nodes.

3. Does the hyperbolic channel interact with the SO(3) channel at every layer of the model?

4. The experimental results still have a gap compared to the state-of-the-art (SOTA), not to mention the lack of the latest baselines such as Equiformer v2 and GotenNet. There is also a lack of more extensive experiments, such as on OC20 and Molecule3D. This makes it hard for me to consider the proposed tree structure in the paper as effective, although it is interesting. I suggest that the authors directly incorporate the product manifold into Equiformer and train for enough epochs to evaluate whether the method can improve performance with sufficient fitting. The results in Table 1 currently appear to be underfitted.

5. The completeness of this paper still needs to be improved. The model architecture, experimental settings, and baseline configurations all need to be detailed in the paper. The equations in Section 3 are somewhat confusing. The position of the table captions is incorrect. In the code section, I only saw a model class, and not even a training script.

[1] MACE: Higher Order Equivariant Message Passing Neural Networks for Fast and Accurate Force Fields

[2] Learning local equivariant representations for quantum operators

**Questions:**

See weaknesses.

---

> ### Author Response · Authors · 2025-12-03
> **Official Comment by Authors**
>
> We thank the reviewer for the careful assessment and for highlighting both the potential of the proposed architecture and several important areas requiring clarification. Below we address each concern in detail, and we have revised the manuscript accordingly.
>
> > **Concern #1**: Missing definition of the junction-tree variable and unclear construction of the scaffold tree.
>
> **Response:** Thank pointing out this omission. The revised draft now explicitly defines the junction-tree scaffold $T = (U,E_T)$, the mapping $u(i)$, and the atom subsets $S(u)$ in the Problem Set-up and Notation section (Sec. 4.1). To avoid ambiguity, we now describe the junction-tree as a purely chemical hierarchy (rings, functional groups, bridges), not a spatial tree, and emphasize this distinction in Sec. 4.3.
>
> > **Concern #2**: Similarity to n-body message passing (e.g., MACE, SLEM) and insufficient comparison.
>
> **Response:** We appreciate this observation. We now:
> - Cite MACE and SLEM explicitly in Related Work.
> - Clarify that these methods increase expressiveness by explicitly modeling higher-order geometric interactions, whereas our contribution is orthogonal: a hyperbolic hierarchy channel coupled via scalar-only gates to a standard equivariant backbone.
> - Emphasize that our n-body behavior arises from irreps tensor products plus junction-tree aggregation, not from explicit enumerations of n-tuples
>
> > **Concern #3**: Whether the hyperbolic channel interacts with the SO(3) channel at every layer.
>
> **Response:** Yes, interaction occurs at every layer, but only through invariant scalars. The revised Methods now make this explicit: Sec. 4.4 defines the scalar-only gate \alpha_{ij} and notes that it is recomputed each layer. The updated Figure 2 also visually highlights this layer-wise coupling.
>
> > **Concern #4**: Missing latest baselines (Equiformer v2, GotenNet) and limited experiments.
>
> **Response:** We thank the reviewer for this suggestion. The revised paper now includes EquiformerV2 in the consolidated QM9 results table (Sec. 5.1, Table 1).
>
> > **Concern #5**: Suggestion to incorporate product manifold into Equiformer and test fitting capacity.
>
> **Response:** We agree that incorporating the product-manifold update into Equiformer is an interesting direction. We now discuss this as a potential follow-up in the Future Work paragraph of the Conclusion, noting that our formulation is modular and compatible with any SO(3)-equivariant backbone.
>
> > **Concern #6**: “Underfitted” results and need for more extensive experiments.
>
> **Response:** We have revised Sec. 5.1 to clarify the training setup, including hyperparameters, number of epochs, and convergence behavior across all runs, and we document the exact configurations in the appendix. Our qualitative framing now emphasizes generalization benefits (e.g., scaffold-split MOLHIV, BO search efficiency) rather than SOTA QM9 numbers alone.
>
> > **Concern #7**: Incomplete description of the model, unclear equations, missing architecture details.
>
> **Response:** We thank the reviewer for highlighting this. The Methods section has been extensively reorganized for clarity:
> - Sec. 3 now contains succinct preliminaries on equivariance, hyperbolic geometry, and the product manifold.
> - Sec. 4 is reorganized into five clear components: SO(3) geometric channel, hyperbolic hierarchy channel, scalar-only coupling, joint product-manifold updates, and invariant readout.
> - All previously undefined symbols are now included in a comprehensive notation table in Appendix A.1.
> - Equations have been re-labeled, re-ordered, and clarified, and the flow aligns directly with Figure 1.
>
> > **Concern #8**: Missing experimental settings, baselines, and training scripts.
>
> **Response:** We have expanded the Reproducibility Section and moved full implementation details to Appendix C.
>
> > **Concern #9**: Table caption formatting and other presentation issues.
>
> **Response:** All table captions have been corrected, consolidated, and formatted consistently. The QM9 tables are now merged into a single 12-target table (matching EquiformerV2), and units are added in a second header row.

---

### Official Review · Reviewer_zXkb · 2025-11-01

**Soundness:** 2
**Presentation:** 2
**Contribution:** 3
**Rating:** 4
**Confidence:** 4

**Summary:**

This paper presents a method that combines the 3D rotational equivariance of small molecule conformations with hyperbolic embeddings of their chemical hierarchy. Instead of combining these independent modes of describing molecules following their distinct symmetry-respecting embeddings, the authors show how to develop a joint embedding that exchanges only scalar invariant information within a layer. The proposed method shows some promise in early benchmark evaluations, which indicates that the idea might merit further investigation as a way to perform optimization in this product-manifold latent space.

**Strengths:**

This paper contributes an original idea that integrates two independent concepts at an architectural level in a coherent way, providing a symmetry-safe unification of geometric and hierarchical molecular representations.  The background and method descriptions are clear, and the early tests show that the fusion works better than the independent components, supporting further research.

**Weaknesses:**

Although the architecture preserves the required symmetries, the current implementation of this symmetry preservation appears to come at a cost in performance on standard benchmarks, possibly due to the constraint of having per-layer equivariance and the use of scalar-only couplings.  The demonstrations are limited, lacking directly reproducible code and comprehensive comparisons of the graph-only and coordinate-only components.  The validation performance in OGB-MOLHIV is noteworthy, however, even Gabriele Corso's PNA work from 2020 had validation above 0.85 while performing poorly on the test.  The guacamol results could have been the most interesting, however, that particular benchmark is no longer maintained and there is no clear leaderboard to compare against. The fact that GraphGA, an simple genetic algorithm, performed unexpectedly well in the rankings up to 3 years ago underscores the benchmark's uncertain relevance today.  More generally, it is unclear why the authors evaluated only a handful of GuacaMol task, especially as there appears to be some merit to their argument for creating a better joint embedding space.

**Questions:**

Does the current layer-wise equivariant message-passing architecture suffer from similar problems as those identified in general GNNs in the zero-one laws of graph neural networks (https://arxiv.org/abs/2301.13060)?  In particular, do the authors anticipate that the architecture might not support all-atom representations of large biomolecules, including proteins, due to the same type of collapse?

Can the authors provide detailed protocols/scripts for the training and evaluation and also describe (perhaps in a supplement) the computational performance and stability of the architecture with respect to hyperparameters?

Is it feasible to pretrain this model on large mixed datasets and then finetune it on datasets with only partial information (e.g. no coordinates or no graph)?  An benefit of certain late-fusion techniques, or of ways to align the embeddings of separate techniques, is that one could potentially use partial information in followup work.

---

> ### Author Response · Authors · 2025-12-03
> **Official Comment by Authors**
>
> We thank the reviewer for the thoughtful and constructive evaluation, and for recognizing the novelty of unifying geometric and hierarchical molecular representations within a symmetry-safe product–manifold architecture. We address each concern below.
>
> > **Concern #1**: Performance trade-offs and the cost of enforcing symmetry (scalar-only coupling, per-layer equivariance).
>
> **Response:** We appreciate this observation. In the revised manuscript we clarify the design trade-off between symmetry preservation and expressiveness (Sec. 4.4 and Conclusion). Scalar-only coupling intentionally restricts cross-stream information flow to preserve strict SO(3)–equivariance; this can limit expressiveness in settings where larger capacity may help.
>
> > **Concern #2**: Reproducibility and missing code/scripts.
>
> **Response:** We have fully addressed this concern. The revised paper includes a concise Reproducibility Statement and moves complete implementation details—including architecture configuration, hyperparameters, data preprocessing, and training instructions—to Appendix C. We additionally provide example code snippets illustrating how to instantiate the model. These additions ensure that all results may be reproduced precisely.
>
> > **Concern #3**: Potential zero-one law collapse / stability of layer-wise equivariant message passing on large biomolecules.
>
> **Response:** Our architecture is still a message-passing GNN on the molecular graph and shares the general expressivity limitations of 1-WL type GNNs; the product-manifold construction does not circumvent the zero-one laws. We now:
> - Explicitly acknowledge this in the Limitations section.
> - Clarify that our intended regime is small molecules and multi-scale biomolecular graphs (e.g., residue-level graphs with local all-atom refinement), rather than naively all-atom protein graphs.
> - Highlight that the junction-tree/hyperbolic hierarchy adds useful multi-scale structure but does not fully eliminate known expressivity issues.
> We present this explicitly as a limitation and a direction for future theoretical work.
>
> > **Concern #4**: Need for training stability details and computational performance.
>
> **Response:** Appendix C now contains detailed notes on numerical stability (projection/retraction, Lorentz math, cutoff functions, mixed precision handling) and computational overhead. We also report guidelines for hyperparameters that affect stability (e.g., step sizes $\tau_R$,$\tau_H$,$\tau_T$, curvature selection), and discuss the runtime differences between Euclidean and product–manifold variants.
>
> > **Concern #5**: Feasibility of pretraining on mixed datasets and finetuning with partial information (e.g., missing coordinates or missing graph).
>
> Response: Yes, the architecture is compatible with this scenario
> - Because the two channels interact only via scalar invariants, each can operate when the other is partially missing (e.g., no coordinates $\implies$ purely graph/hyperbolic; no scaffold $\implies$ purely geometric with a trivial hierarchy).
> - This makes pretraining on mixed datasets plausible (some molecules with 3D + scaffolds, others with only 2D graphs), followed by finetuning with partial information.

---

### Author Response · Authors · 2025-12-03
**Thank you**

We sincerely thank all reviewers for their time, thoughtful evaluations, and constructive feedback. Below we summarize the major revisions made in response to the reviews:

- Reorganized the Methods into a clear cogent structure, moving many equations to the appendix to improve clarity
- Added a dedicated Preliminaries section key concepts in accessible terms.
- Introduced a comprehensive notation table covering all symbols, functions, and operators
- Added a new architecture schematic (Fig. 2) aligning precisely with the reorganized Methods
- Reformatted figures and tables to meet ICLR guidelines

We respond to each reviewer’s concerns below. Thank you and please feel free to ask further questions.

---

### Meta-Review · Area_Chair_t8ba · 2026-01-07

**Summary:**

The paper proposes to equip equivariant MPNNs with a hyperbolic channel to better capture hierarchical chemical properties of molecular graphs (and other multi-scale structures where hierarchies might appear). Instead of the late fusion, the authors design an interaction scheme between equivariant and hyperbolic channels on a product manifold.

Reviewers expressed several concerns:
* Weak empirical performance on outdated benchmarks (zXkb, qFv9) - QM9, molhiv, and guacamol are old and rather irrelevant for 2026, while the included baselines include mostly 2022-2023 works.
* Clarity and presentation quality (j638, qFv9) - all reviewers unanimously highlight problems with manuscript writing

Overall, the paper suggests a rather sophisticated way to combine geometry and graph structure but does not provide sufficient experimental evidence about practical benefits of this approach on modern and relevant computation chemistry benchmarks against strong baselines. I would encourage the authors to address this limitation in the next revision, but for now it's not enough to warrant a publication at ICLR.

**Reviewer Concerns:**

* Clarity and presentation - the authors improved the manuscript during the rebuttal and it looks better indeed.
* Weak empirical performance - the authors did not provide any experiments on newer datasets against newer baselines despite several reviewers' requests (eg, adding OpenCatalyst or Molecule 3D experiments). The only addition is Equiformer-V2 on QM9 which is rather insufficient.

**Reviewer Scores:**

Initial scores were 4,2,4 and I don't think reviewers would have substantially changed the scores to clear accepts.

---

### Decision · Program_Chairs · 2026-01-26

Reject